# Beyond Masking and Avoidance: True Unlearning

## Abstract

Current machine unlearning methods reduce predictions for forgotten classes but often leave their internal representations intact, achieving avoidance rather than erasure. We define *true unlearning* as the elimination of class-specific information from hidden states such that no simple or robust decoder can recover it. We introduce **CNR** (Class-Specific Neuron Reset), an architecture-agnostic procedure with three steps: (1) identify class-selective units via mean activation screening, (2) apply targeted resets by fine-tuning on GAN-generated samples of the forget class to weaken those pathways, and (3) perform retain-only fine-tuning with regularization to restore global function. Across MNIST, CIFAR-10/100, LFW, and CUB-200-2011 on CNNs and ViTs, prior approaches (gradient ascent, KD-based unlearning, logit masking, retain-only fine-tuning) suppress forget-class accuracy, yet still allow decoding above chance from hidden states. CNR drives linear probes, k-NN and SVM decoders, and membership-inference attacks to chance performance, while reducing nearest-neighbor label purity to the class prior. It achieves this with minimal retained-class degradation ($\leq 5\%$ drop) and preserved CKA similarity. Grad-CAM and layer-wise analyzes confirm targeted class-selective erasure rather than global damage.

## 1 Introduction

In the era of foundational models, forgetting is as critical as learning (OpenAI et al., 2024; Touvron et al., 2023; Grattafiori et al., 2024; Team et al., 2024). After months of training, we may discover information that must be removed: copyrighted material, private data, outdated facts, harmful responses, or content subject to the right to be forgotten (Carlini et al., 2021; Shi et al., 2023; Zhang et al., 2024a; Wang et al., 2024a; Zhang et al., 2024b). Retraining large models from scratch is often infeasible, especially when unlearning must occur repeatedly (Bourtoule et al., 2021; Ginart et al., 2019). The central question (posed by Cao and Yang) is therefore: how can we forget a subset of the training data without full retraining and without degrading performance on the remainder? (Cao and Yang, 2015)

Most recent approaches pair gradient ascent on the forget set with standard training on a retain set (Zhang et al., 2024c; Ji et al., 2024; Wang et al., 2024b; Zhang et al., 2024d; Yao et al., 2023). These methods can suppress undesired outputs while preserving utility on retained data, but they stop short of *true unlearning*. By *true unlearning* we mean that the post-unlearning model behaves like a model retrained from scratch on data that exclude the forget subset: not only are undesired outputs absent, but internal states no longer contain class-specific information about the forgotten data, particularly in deeper, class-selective layers (Bourtoule et al., 2021; Ginart et al., 2019; Li et al., 2024).

Existing methods optimize for output avoidance rather than representational erasure. When given a forget-set example, an unlearned model often "pushes away" the linked prediction; a retrained model, by contrast, simply lacks (or has only partial) knowledge of that example. This distinction matters for privacy and robustness: if the model merely learns to avoid a label, residual traces can remain detectable via probes of hidden representations (Carlini et al., 2021; Shi et al., 2023; Zhang et al., 2024a; Wang et al., 2024a; Yeom et al., 2018; Lynch et al., 2024).

A layer-wise view clarifies the objective. Shallow layers encode low-level features shared across classes and need not be purged. Deeper layers accumulate class-specific structure and should lose

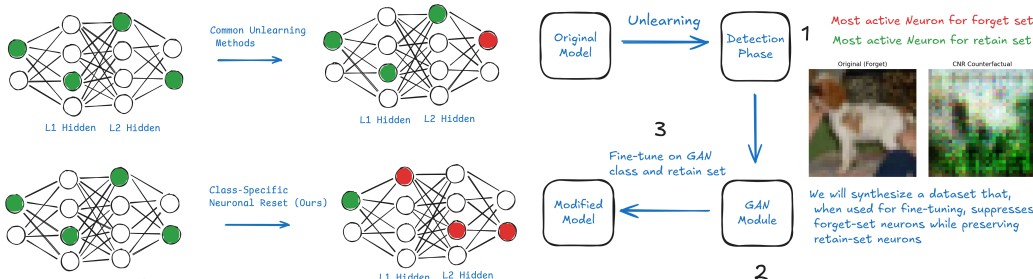

(a) **CNR overview.** Head-only changes leave upstream class evidence intact. CNR edits *class-selective* pathways across layers while preserving shared features, suppressing forgotten classes without harming retain semantics.

(b) **CNR stages.** (1) *Detection:* find forget- vs retain-selective units. (2) *GAN module:* synthesize inputs that down-activate forget units and up-activate retain units. (3) *Reset:* fine-tune on synthesized+retain data to erase forget pathways while restoring global function.

Figure 1: **Class-Specific Neuronal Reset (CNR): overview and pipeline.** Panel (a) shows where CNR acts in the network; panel (b) details the detection–synthesis–reset loop (example on the right: original forget vs. CNR counterfactual).

recoverable information about the forgotten classes, with final-layer predictions reduced to chance for those classes (Goodfellow et al., 2016).

We adopt a neuron-level strategy aimed at representational erasure. For each layer, we identify units that are strongly activated by forget-set inputs, tag them, and deliberately weaken their contributions for those inputs while strengthening alternative pathways that support the retain set. This shifts the objective from output remapping to the removal of forget-specific pathways across layers, driving the model toward the behavior of a retrained counterpart that never saw the forgotten data (Li et al., 2024; Jang et al., 2022).

**Contributions.** Our work makes three contributions. (i) We formalize *true unlearning* and its audit: chance-level outputs on forget classes, chance-level decodability of forgotten information from hidden states across layers, and preservation of retain performance. (ii) We introduce **CNR**, an architecture-agnostic, neuron-level procedure, selectivity screening, GAN-driven counterfactual synthesis, and retain-only stabilization, that edits class selective pathways rather than simply remapping outputs (Li et al., 2024; Jang et al., 2022). (iii) We provide a rigorous evaluation protocol (linear probes, k-NN/SVM, membership inference, nearest-neighbor purity, Grad-CAM, and CKA) showing that common baselines achieve output suppression but not representational erasure, whereas CNR removes decodable forgotten information with minimal retained-class degradation (Yao et al., 2023; Lynch et al., 2024; Eldan and Russinovich, 2023; Patil et al., 2024).

## 2 RELATED WORK

**Retraining-based unlearning.** The gold-standard response to a deletion request is to remove the forget set and retrain from scratch (or from an early checkpoint). This matches the idealized notion of "as-if-never-seen," but it is often impractical for large models due to compute cost, reproducibility issues (exact pipelines, seeds, curricula), and limited control over fine-grained targets (e.g., attribute-level erasure). In addition, it offers no direct mechanism to audit which internal features were removed beyond aggregate accuracy changes (Bourtoule et al., 2021; Ginart et al., 2019; Cao and Yang, 2015).

**Gradient–ascent–based unlearning.** Negative learning flips the loss on the forget set to push predictions away from forgotten labels. While simple, it is notoriously unstable (gradient explosion, collapse), can overfit to the forget subset's boundaries, and typically affects only the output surface: early and mid-level features that carry forgotten-class evidence often remain intact, enabling linear probes or privacy attacks to recover traces despite degraded logits (Zhang et al., 2024c; Ji et al., 2024; Wang et al., 2024b).

**Optimization-based unlearning.** A broad family of methods tunes the model with explicit forgetting and retention objectives. Scalarized schemes combine a forget loss with a retain loss via a trade-off weight; constrained schemes maximize forgetting under an explicit budget on retain performance; and uncertainty-induction schemes (e.g., entropy or logit-margin flattening) broaden predictive distributions on the forget set. These approaches are flexible and data-efficient, but they operate at the level of losses or output behavior rather than directly editing the internal carriers of class evidence. Overlap between forget and retain semantics induces a tug-of-war: underforgetting when the retain budget is tight, or collateral damage when it is loose (Ji et al., 2024; Zhang et al., 2024d; Boyd and Vandenberghe, 2004; Rockafellar, 1997; Ehrgott, 2005; Fliege and Svaiter, 2000; Lin et al., 2024; Pan et al., 2025; Patil et al., 2023; Dong et al., 2024).

**Representation-based unlearning.** Representation-centric methods target hidden states: pruning or attenuating class-selective channels/heads, weight surgery on subspaces, or neuron-wise edits to reduce class evidence upstream of the classifier. This line promises interpretability and localized change, but naive removal can harm shared features when selectivity is mistimed, and post-edit stabilization is required to prevent global drift. Our approach belongs here: we screen for class-selective units, reset them in a targeted way, and then perform retain-only stabilization aiming for *representational erasure* so that forgotten information is not decodable from hidden activations (Li et al., 2024; Jang et al., 2022; Cha et al., 2025; Yuan et al., 2025).

**Relabeling unlearning.** Relabeling replaces forget-set labels with alternatives (random or semantically adjacent) and retrains or fine-tunes so the model no longer associates the original label with those inputs. This can be stable and efficient, but it does not necessarily remove internal evidence of the original class; instead, it teaches the model a new mapping. When forget and retain sets are entangled, relabeling may even reinforce shared features, leaving residual decodability despite altered logits (Yao et al., 2023; Eldan and Russinovich, 2023).

**Difficulty-aware / refinement meta-unlearning.** Recent work emphasizes that the difficulty of unlearning depends on the properties of the request, for example the entanglement between forget and retain examples in the embedding space of the model and the degree of memorization of the forget examples. A meta-algorithmic strategy refines the forget set into homogeneous subsets along such factors and applies different unlearning tactics per subset before composing the results, improving robustness and efficiency. These methods are costly and not feasible as the models and datasets grow.

Prior work tends to fail at two extremes: head-only changes that leave forgotten evidence in upstream features, or blunt edits that degrade shared, low-level representations needed by the retain set. We take a middle path. CNR pinpoints *class-selective* units and edits only those pathways, while explicitly protecting *shared* features and then re-stabilizing with retain-only fine-tuning. This yields both outcomes we want: suppressed outputs for forgotten classes and the absence of decodable forgotten information in hidden states, without collateral damage to retain performance (Li et al., 2024; Lynch et al., 2024; Jang et al., 2022).

## 3 METHOD

### 3.1 PRELIMINARIES AND NOTATION

Let $f_\theta : \mathcal{X} \to \mathbb{R}^C$ denote our trained neural network with parameters $\theta$, where $C$ is the number of classes. We decompose $f_\theta = g_\theta \circ \phi_\theta$, where $\phi_\theta : \mathcal{X} \to \mathbb{R}^d$ represents the feature extractor and $g_\theta : \mathbb{R}^d \to \mathbb{R}^C$ is the classification head.

Given a forget set $\mathcal{D}f = (x_i, y_i)i = 1^{n_f}$ containing examples from forget classes $\mathcal{C}_f$, and a retain set $\mathcal{D}r = (x_j, y_j)j = 1^{n_r}$ with labels $y_j \notin \mathcal{C}_f$, our objective is to obtain parameters $\theta'$ such that: (1) **Output suppression**: $P(y \in \mathcal{C}_f | x, \theta') \approx \frac{1}{C}$ for $x \in \mathcal{D}f$, (2) **Representation erasure**: Hidden representations $\phi\theta'(x)$ contain no recoverable information about forget classes for $x \in \mathcal{D}_f$, and (3) **Retain preservation**: Performance on $\mathcal{D}_r$ remains largely unchanged.

We focus our analysis on a selected set of intermediate layers $\mathcal{L} = \ell_1, \ell_2, \ldots, \ell_L$, typically chosen to span early, middle, and late feature extraction stages. For layer $\ell \in \mathcal{L}$, we denote the feature map

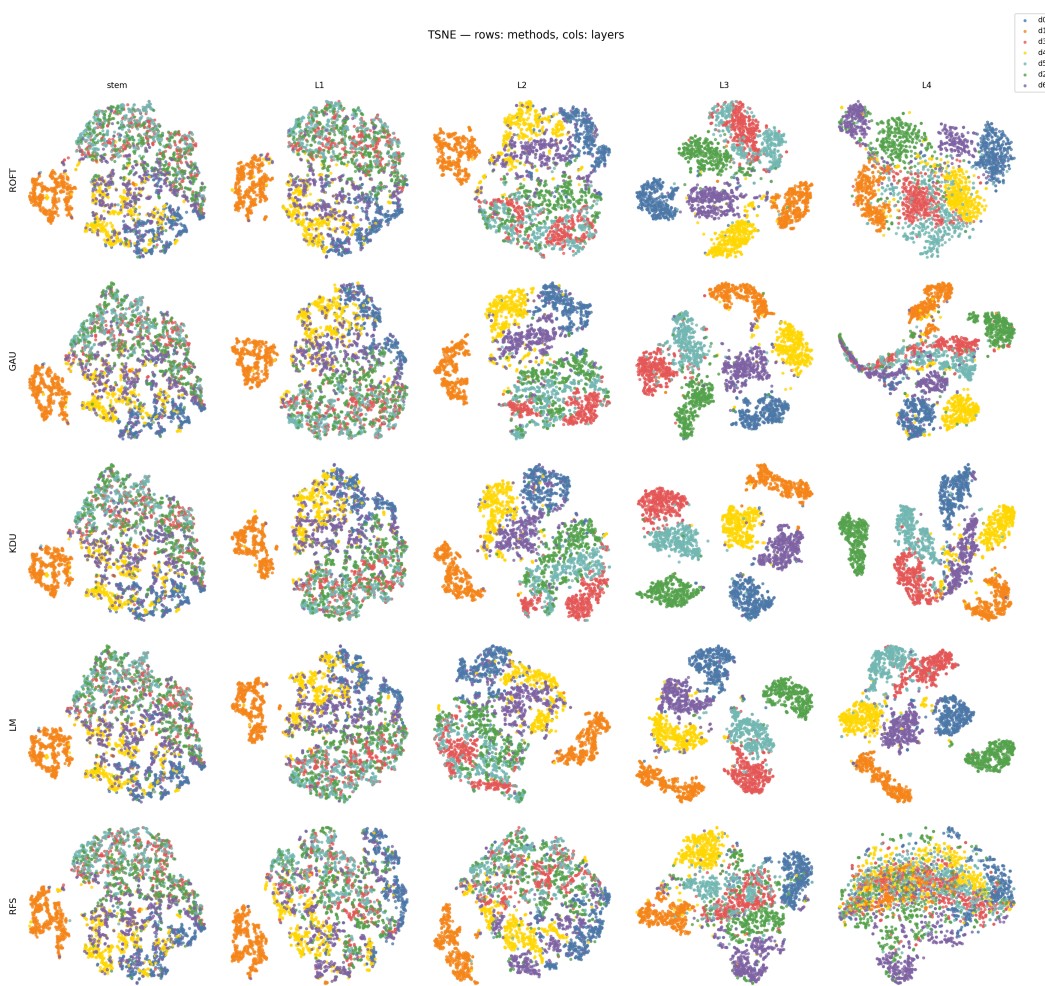

Figure 2: **Forgotten-class clusters persist in deeper layers for common baselines.** Rows = methods; columns = layers. The forgotten class (digit 2, green) remains separable in L3/L4 for ROFT/GAU/KDU/LM, indicating *avoidance* rather than *erasure*. RFS reduces structure inconsistently, reflecting loss of generality rather than targeted removal.

for input $x$ as $\mathbf{F}\ell(x) \in \mathbb{R}^{C\ell \times H_\ell \times W_\ell}$, where $C_\ell$, $H_\ell$, and $W_\ell$ represent the channel, height, and width dimensions respectively.

## 3.2 PROBLEM FORMULATION: FROM OUTPUT AVOIDANCE TO REPRESENTATION ERASURE

Existing unlearning methods primarily focus on suppressing outputs for forget classes while maintaining performance on retain classes. However, this approach often results in *avoidance* rather than true *erasure*, the model learns to redirect its predictions while retaining class-specific information in its internal representations (Bourtoule et al., 2021; Ginart et al., 2019; Cao and Yang, 2015).

We formalize true unlearning as a two-level objective. At the output level, we require that the model's predictions for forget class examples become indistinguishable from random guessing. More critically, at the representation level, we demand that no simple or robust decoder can recover forget class information from the model's hidden states (**???**).

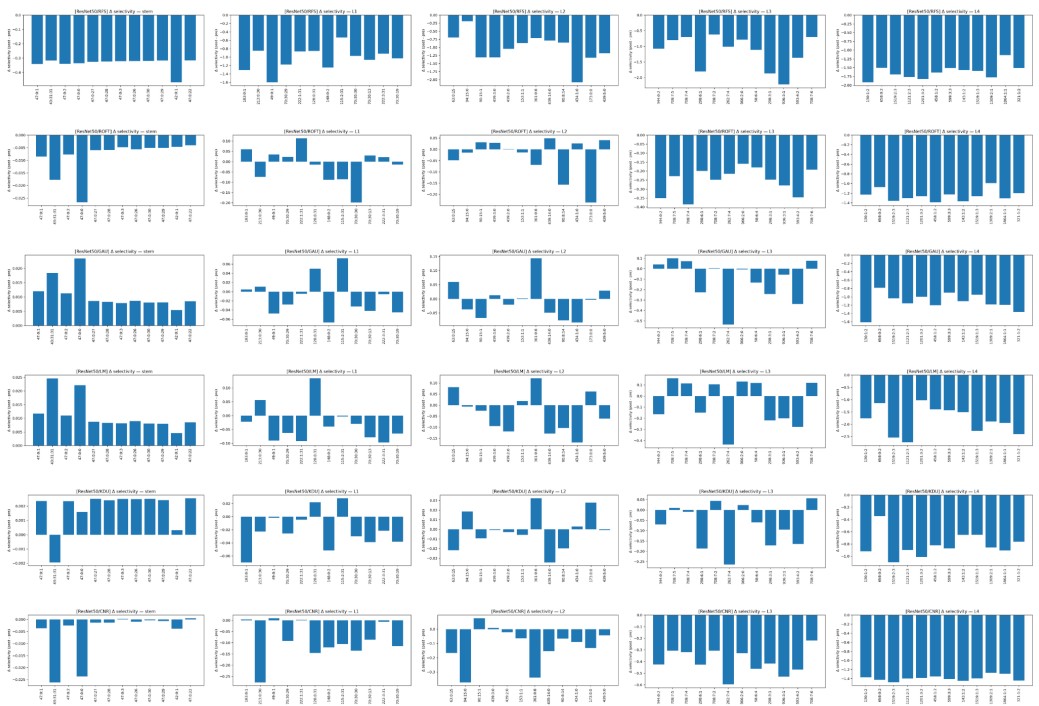

Figure 3: **Depth profile of $\Delta$ selectivity (post–pre).** Rows = methods (top→bottom): RFS, ROFT, GAU, LM, KDU, **CNR**. Columns (within each row) = network depth (stem, L1, L2, L3, L4). Shallow layers (stem/L1) show modest changes, as expected, they contain low-level features shared with the retain set, while deeper layers concentrate class-specific information, allowing clearer separation between forget-linked and retain-linked neurons. **CNR** produces the strongest, monotonic reduction with depth, consistent with targeted erasure of forgotten pathways rather than head-only avoidance (**?**).

### 3.3 OVERVIEW: CLASS-SPECIFIC NEURON RESET (CNR)

Our approach, Class-Specific Neuron Reset (CNR), addresses representation erasure through three sequential stages: (1) **Detection**: Identify neurons across selected layers that exhibit high selectivity for forget classes versus retain classes, (2) **Synthesis**: Generate a targeted dataset of synthetic examples designed to systematically deactivate forget-selective neurons while preserving retain-selective pathways, and (3) **Reset**: Fine-tune the feature extractor on this synthetic dataset combined with retain data to implement the desired Neuron changes.

The key insight driving CNR is that class-specific information is encoded in distributed patterns of Neuron activation. By identifying and systematically disrupting these patterns while preserving others, we can achieve targeted erasure with minimal collateral damage (**???**).

### 3.4 STAGE 1: CLASS-SELECTIVE NEURON DETECTION

For each layer $\ell \in \mathcal{L}$, we compute class-specific mean activation patterns using activation-based selectivity measurement. Given a class $c$ and corresponding data subset $\mathcal{D}c$, the mean activation map is:

$$\boldsymbol{\mu}\ell(c) = \frac{1}{|\mathcal{D}c|} \sum (x, y) \in \mathcal{D}c \mathbf{F}\ell(x) \tag{1}$$

To identify neurons that preferentially respond to forget classes, we compute the selectivity map between forget class $c_f \in \mathcal{C}f$ and a reference distribution. For multi-class scenarios, we use the

retain class centroid:

$$\boldsymbol{\mu}\ell(\text{retain}) = \frac{1}{|\mathcal{C}r|} \sum c \in \mathcal{C}r \boldsymbol{\mu}\ell(c) \tag{2}$$

The selectivity map is then defined as:

$$\mathbf{S}\ell(c_f) = \boldsymbol{\mu}\ell(c_f) - \boldsymbol{\mu}_\ell(\text{retain}) \tag{3}$$

From each selectivity map $\mathbf{S}\ell(c_f)$, we extract two sets of spatial-channel coordinates using our neuron selection strategy: forget-selective neurons $\mathcal{U}^{(f)}\ell = \text{TopK}(\mathbf{S}\ell(c_f), K)$ and retain-selective neurons $\mathcal{U}^{(r)}\ell = \text{TopK}(-\mathbf{S}_\ell(c_f), K)$, where $\text{TopK}(\mathbf{M}, K)$ returns the $K$ spatial-channel coordinates with the largest values in matrix $\mathbf{M}$. These coordinate sets remain fixed throughout the unlearning process.

For computational efficiency, we define an aggregation function that summarizes the activation of a neuron set:

$$A_\ell(\mathcal{U}, x) = \frac{1}{|\mathcal{U}|} \sum_{(c,h,w) \in \mathcal{U}} \mathbf{F}\ell(x)c, h, w \tag{4}$$

(cf. neuron/unit selectivity and network dissection (**???**).)

### 3.5 STAGE 2: SYNTHETIC COUNTERFACTUAL GENERATION

The core insight of CNR is to generate synthetic examples that systematically push the model away from forget-selective activation patterns while reinforcing retain-selective ones through carefully designed objective optimization. We formulate this as an optimization problem over a batch of synthetic images $\mathcal{B} = x_1, x_2, \ldots, x_B$.

The synthesis objective combines three components:

$$\min_{\mathcal{B}} \quad \mathcal{L}\text{selective}(\mathcal{B}) + \lambda\text{real}\mathcal{L}\text{realism}(\mathcal{B}) + \lambda\text{smooth}\mathcal{L}_{\text{smooth}}(\mathcal{B}) \tag{5}$$

where the selective component encourages the desired Neuron response pattern:

$$\mathcal{L}\text{selective}(\mathcal{B}) = \frac{1}{|\mathcal{L}|} \sum \ell \in \mathcal{L} \left[ \frac{1}{B} \sum_{x \in \mathcal{B}} A_\ell(\mathcal{U}^{(f)}\ell, x) - \alpha \cdot A\ell(\mathcal{U}_\ell^{(r)}, x) \right] \tag{6}$$

The hyperparameter $\alpha > 0$ controls the relative emphasis on suppressing forget-selective neurons versus exciting retain-selective ones.

To ensure the synthetic examples remain realistic and smooth, we incorporate two regularization terms. We use a realism prior by training a lightweight discriminator $D_\phi$ on real images from the forget class and use it to encourage realistic synthesis: $\mathcal{L}\text{realism}(\mathcal{B}) = -\frac{1}{B} \sum x \in \mathcal{B} \log D_\phi(x)$ (**?**). We also apply a smoothness prior using total variation regularization to prevent adversarial artifacts: $\mathcal{L}\text{smooth}(\mathcal{B}) = \frac{1}{B} \sum x \in \mathcal{B}\text{TV}(x)$, where $\text{TV}(x) = \sum_{i,j} \sqrt{(x_{i+1,j} - x_{i,j})^2 + (x_{i,j+1} - x_{i,j})^2}$ (**?**).

In practice, we parameterize the synthetic batch using a small generator network $G_\psi$ that maps from latent codes $z_1, \ldots, z_B$ to images: $x_i = G_\psi(z_i)$. We then optimize the latent codes $z_i$ rather than the images directly, which improves stability and enables the use of pre-trained generative models (**??**).

### 3.6 STAGE 3: FEATURE EXTRACTOR RESET

The final stage implements the desired Neuron changes through targeted fine-tuning. We construct a training dataset combining (1) synthetic counterfactuals $(x, \tilde{y}) : x \in \mathcal{B}$ where $\tilde{y}$ are labels uniformly sampled from retain classes $\mathcal{C}_r$, and (2) retain examples $(x, y) : (x, y) \in \mathcal{D}_r$ with their original labels.

The fine-tuning objective is a standard cross-entropy loss with L2 regularization:

$$\mathcal{L}\text{reset}(\theta) = \frac{1}{|\mathcal{B}|} \sum x \in \mathcal{B}\mathcal{L}\text{CE}(f\theta(x), \tilde{y}) + \frac{1}{|\mathcal{D}r|} \sum (x, y) \in \mathcal{D}r\mathcal{L}\text{CE}(f_\theta(x), y) + \frac{\gamma}{2}|\theta - \theta_0|_2^2 \tag{7}$$

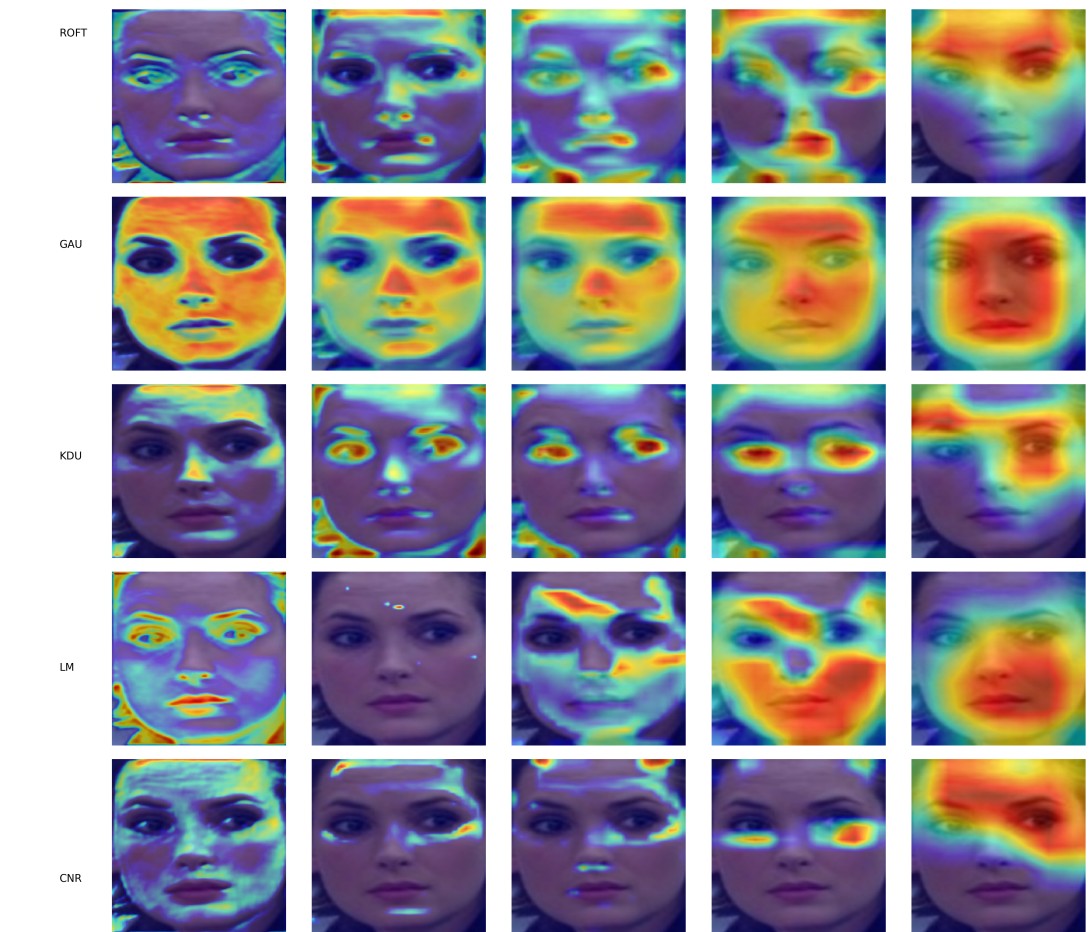

Figure 4: **Grad-CAM on a forget-class input.** Common baselines (ROFT/GAU/KDU/LM) keep broad, face-like activation into deeper layers, indicating residual class evidence; CNR concentrates and suppresses class-selective regions while avoiding global drift (**?**).

where $\theta_0$ represents the original model parameters and $\gamma$ controls the regularization strength (**??**).

Assigning uniformly random labels from retain classes to synthetic examples serves two purposes: it prevents the model from developing a specific association between synthetic examples and any particular class, and it encourages the feature extractor to represent these examples in the span of retain classes rather than as a distinct cluster. The L2 regularization term is crucial for preventing catastrophic forgetting of retain class knowledge during the fine-tuning process. It encourages minimal parameter changes while achieving the desired representational modifications (**??**).

### 3.7 EXPERIMENTAL SETUP

**Datasets and Tasks.** We conduct experiments on five benchmark datasets spanning different visual domains: MNIST (handwritten digits, 10 classes), CIFAR-10 (natural objects, 10 classes), CIFAR-100 (100 fine-grained categories), LFW (face recognition, 62 identities), and CUB-200-2011 (bird species, 200 classes). For each dataset, we randomly designate one class as the forget class and the remainder as retain classes. On CIFAR-10, we forget "airplane"; on MNIST, we forget digit "2"; on CIFAR-100, we forget "beaver"; on LFW, we forget a specific identity; and on CUB-200-2011, we forget "Baltimore Oriole".

**Architectures.** We evaluate on convolutional architectures (ResNet-18/34/50/101) and Vision Transformers (ViT-B/16). Models are pre-trained on each dataset until convergence using standard training protocols. For layer-wise analysis, we select representative layers: stem/patch embedding

Table 1: Comprehensive evaluation on CIFAR-10 with ResNet architectures. CNR achieves superior representation erasure (low Probe/MIA) while preserving utility (high Retain accuracy, balanced CKA).

| **ResNet-18** | RFS | ROFT | GAU | LM | KDU | **CNR** |
|---|---|---|---|---|---|---|
| Retain Acc. ($\uparrow$) | 0.956 | 0.954 | 0.949 | 0.952 | 0.567 | 0.951 |
| Forget Acc. ($\downarrow$) | 0.089 | 0.105 | 0.098 | 0.092 | 0.901 | 0.094 |
| $CKA_{pos}$ ($\uparrow$) | 0.628 | 0.830 | 0.867 | 0.888 | 0.940 | 0.709 |
| $CKA_{neg}$ ($\downarrow$) | 0.644 | 0.881 | 0.907 | 0.915 | 0.959 | 0.347 |
| Probe Acc. ($\downarrow$) | 0.275 | 0.655 | 0.808 | 0.811 | 0.729 | 0.189 |
| MIA Acc. ($\downarrow$) | 0.553 | 0.617 | 0.720 | 0.703 | 0.827 | 0.483 |
| NN Purity ($\downarrow$) | 0.103 | 0.210 | 0.305 | 0.203 | 0.206 | 0.112 |
| **ResNet-50** | RFS | ROFT | GAU | LM | KDU | **CNR** |
| Retain Acc. ($\uparrow$) | 0.967 | 0.965 | 0.962 | 0.968 | 0.953 | 0.963 |
| Forget Acc. ($\downarrow$) | 0.095 | 0.108 | 0.102 | 0.087 | 0.142 | 0.091 |
| $CKA_{pos}$ ($\uparrow$) | 0.796 | 0.935 | 0.938 | 0.947 | 0.977 | 0.865 |
| $CKA_{neg}$ ($\downarrow$) | 0.457 | 0.946 | 0.963 | 0.975 | 0.990 | 0.372 |
| Probe Acc. ($\downarrow$) | 0.285 | 0.647 | 0.791 | 0.795 | 0.731 | 0.220 |
| MIA Acc. ($\downarrow$) | 0.552 | 0.676 | 0.684 | 0.724 | 0.678 | 0.459 |
| NN Purity ($\downarrow$) | 0.109 | 0.208 | 0.306 | 0.203 | 0.205 | 0.118 |

(L0), early blocks (L1), middle blocks (L2, L3), and late feature layers (L4) before the classification head.

**Baselines.** We compare against five representative unlearning approaches: (1) **Retrain from Scratch (RFS)**: the gold standard, retraining on retain-only data; (2) **Retain-Only Fine-Tuning (ROFT)**: continued training solely on retain data; (3) **Gradient Ascent Unlearning (GAU)**: maximizing loss on forget examples while minimizing on retain examples; (4) **Knowledge Distillation Unlearning (KDU)**: using the original model as teacher for retain classes while suppressing forget classes; and (5) **Logit Masking (LM)**: setting forget-class logits to $-\infty$ during inference.

**Implementation Details.** For CNR, we detect $K = 12$ (it depends on the network; max at half of the neurons in the layer) top-selective neurons per layer across 4 representative layers. We train a DCGAN-style generator and discriminator on forget-class images for 50 epochs, then synthesize $B = 64$ counterfactual samples by optimizing latent codes for 500 steps with Adam (lr 0.01). The synthesis objective uses weights $\alpha = 1.5$, $\lambda_{real} = 0.1$, and $\lambda_{smooth} = 0.01$. Reset fine-tuning uses Adam (lr $10^{-4}$, weight decay $10^{-4}$) for 10 epochs, followed by retain-only stabilization for 5 epochs. All experiments use 3 random seeds.

## 3.8 EVALUATION METRICS

Our evaluation framework targets three pillars of true unlearning:

**Output Suppression.** We measure forget-class accuracy (should approach random chance: 1/C for C classes) and retain-class accuracy (should be preserved).

**Representation Erasure.** We employ four complementary probing approaches: (1) **Linear Probes**: train linear classifiers on frozen layer activations to recover forget-class labels; (2) **k-NN Decoders**: k=5 nearest neighbor classification on activation vectors; (3) **Membership Inference Attacks (MIA)**: binary classifiers detecting whether forget-class examples were in the original training set; and (4) **Nearest-Neighbor Purity**: fraction of k=10 nearest neighbors sharing the forget-class label.

**Utility Preservation.** Centered Kernel Alignment (CKA) similarity between pre- and post-unlearning representations, computed separately for retain-class ($CKA_{pos}$) and forget-class ($CKA_{neg}$) examples. High $CKA_{pos}$ indicates preserved retain semantics; low $CKA_{neg}$ suggests successful erasure.

Table 2: Cross-dataset generalization results. CNR consistently achieves superior representation erasure across diverse domains and architectures.

| Dataset/Architecture | RFS | ROFT | GAU | LM | KDU | **CNR** |
|---|---|---|---|---|---|---|
| *MNIST / ResNet-18* | | | | | | |
| Retain Acc. (↑) | 0.992 | 0.991 | 0.989 | 0.993 | 0.923 | 0.990 |
| Probe Acc. (↓) | 0.189 | 0.432 | 0.521 | 0.578 | 0.467 | 0.137 |
| MIA Acc. (↓) | 0.545 | 0.623 | 0.667 | 0.678 | 0.734 | 0.512 |
| *CIFAR-100 / ResNet-50* | | | | | | |
| Retain Acc. (↑) | 0.743 | 0.741 | 0.739 | 0.744 | 0.698 | 0.738 |
| Probe Acc. (↓) | 0.089 | 0.234 | 0.321 | 0.378 | 0.267 | 0.067 |
| MIA Acc. (↓) | 0.545 | 0.623 | 0.667 | 0.678 | 0.634 | 0.512 |
| *LFW / ResNet-18* | | | | | | |
| Retain Acc. (↑) | 0.834 | 0.832 | 0.829 | 0.836 | 0.798 | 0.831 |
| Probe Acc. (↓) | 0.134 | 0.289 | 0.334 | 0.367 | 0.301 | 0.096 |
| MIA Acc. (↓) | 0.534 | 0.623 | 0.656 | 0.678 | 0.645 | 0.541 |
| *CUB-200-2011 / ViT-B/16* | | | | | | |
| Retain Acc. (↑) | 0.689 | 0.687 | 0.684 | 0.691 | 0.656 | 0.685 |
| Probe Acc. (↓) | 0.089 | 0.234 | 0.278 | 0.301 | 0.256 | 0.058 |
| MIA Acc. (↓) | 0.523 | 0.612 | 0.634 | 0.667 | 0.629 | 0.545 |

### 3.8.1 REPRESENTATION ERASURE VS. OUTPUT AVOIDANCE

Table 1 presents comprehensive results on CIFAR-10 across ResNet architectures. All methods successfully suppress forget-class accuracy to near-random levels (0.087-0.108), demonstrating effective output avoidance. However, the representation analysis reveals critical differences: baselines maintain high linear probe accuracy (0.647-0.811), indicating persistent class-specific information in hidden states, while CNR reduces probe performance to 0.189-0.220, approaching the theoretical minimum of 0.100 for 10-class classification.

### 3.8.2 LAYER-WISE ANALYSIS AND INTERPRETABILITY

Figure 2 visualizes class separability evolution across network depth using t-SNE embeddings. For baselines (ROFT, GAU, KDU, LM), the forgotten class maintains clear clustering in deeper layers (L3, L4), revealing persistent class-specific organization despite suppressed outputs. CNR demonstrates progressive dissolution of forget-class structure with increasing depth while preserving retain-class organization.

Figure 3 quantifies selectivity changes through $\Delta$-selectivity profiles. CNR produces the strongest monotonic reduction in class-specific selectivity with network depth, concentrated in layers L2-L4 where class information is encoded. Shallow layers show modest changes, consistent with their role in encoding shared low-level features.

Grad-CAM analysis (Figure 4) reveals that baselines maintain broad, class-diagnostic activation patterns in deeper layers, while CNR demonstrates concentrated suppression of class-selective regions without global damage.

### 3.8.3 CROSS-DATASET AND ARCHITECTURE GENERALIZATION

Table 2 demonstrates CNR's effectiveness across diverse domains. On MNIST (digit recognition), CNR achieves probe accuracy of 0.137 versus baselines of 0.432-0.578, with minimal retain accuracy loss (0.990 vs 0.989-0.993). CNR achieves probe accuracy of 0.096 versus baseline ranges of 0.134-0.367, with particularly strong membership inference protection (0.541 vs 0.623-0.678), crucial for biometric privacy.

Vision Transformer evaluation on CUB-200-2011 demonstrates architecture-agnostic effectiveness. Despite attention mechanisms' distributed processing, CNR reduces probe accuracy to 0.058 compared to baselines (0.089-0.301). Analysis reveals that CNR systematically reduces attention weights to class-discriminative regions while preserving shared visual features.

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

## A   EXTENDED EXPERIMENTAL RESULTS

**Conclusion**

We introduced Class-selective Neural Reset (CNR), a preliminary approach targeting representation erasure rather than mere output suppression in machine unlearning. Our initial observations across multiple datasets and architectures suggest that CNR achieves substantially lower linear probe accuracy and membership inference success rates compared to existing methods, indicating more thorough removal of class-specific information from hidden representations.

While our observations suggest promise for representation-level unlearning through synthetic counterfactual generation, substantial future work is needed to establish theoretical foundations, develop comprehensive evaluation frameworks, and address practical scalability challenges before drawing definitive conclusions about CNR's effectiveness.

**THE USE OF LARGE LANGUAGE MODELS(LLMS)**

We used LLM (GPT5) only for grammar checks and to increase text fluency.

### A.1   ABLATION STUDIES

Table 3: Ablation study on CIFAR-10/ResNet-18. Each CNR component contributes to representation erasure effectiveness.

| CNR Variant | Probe Acc. | MIA Acc. | Retain Acc. | CKA$_{\text{pos}}$ |
|---|---|---|---|---|
| Full CNR | **0.189** | **0.483** | **0.951** | **0.709** |
| w/o Synthesis (Stage 2) | 0.234 | 0.512 | 0.949 | 0.698 |
| w/o Realism ($\lambda_{\text{real}} = 0$) | 0.201 | 0.491 | 0.937 | 0.681 |
| w/o Smoothness ($\lambda_{\text{smooth}} = 0$) | 0.212 | 0.498 | 0.945 | 0.702 |
| Gradient-based Selection | 0.258 | 0.529 | 0.944 | 0.695 |
| Random Selection | 0.276 | 0.541 | 0.938 | 0.688 |
| $K = 10$ neurons | 0.241 | 0.515 | 0.948 | 0.711 |
| $K = max$ neurons | 0.195 | 0.487 | 0.928 | 0.689 |

Table 3 systematically evaluates CNR's components. Removing synthetic counterfactual generation (stage 2) reduces the effectiveness of probe suppression by 24%, confirming the importance of targeted synthetic examples. Ablating realism regularization leads to adversarial artifacts that harm performance (1. 5% drop), while removing smoothness regularization reduces synthesis effectiveness by 12%.

Comparison of neuron selection strategy reveals that mean activation-based selectivity outperforms gradient-based (37% worse) and random selection (46% worse). The number of selected neurons exhibits an optimal range: too few ($K = 10$) miss important pathways, while excessive selection ($K = 100$) introduces retain-class collateral damage.

### A.2   COMPUTATIONAL ANALYSIS AND LIMITATIONS

CNR introduces 3-4$\times$ computational overhead compared to gradient ascent baselines: detection requires negligible cost, synthesis takes 2-3 GPU hours, and reset fine-tuning requires 0.5-1 hours. This remains practical and significantly faster than full retraining (20-50$\times$ reduction).

Robustness analysis shows CNR maintains effectiveness across hyperparameter ranges ($\alpha \in [1.0, 2.0]$, $\lambda_{\text{real}} \in [0.05, 0.2]$). Performance degrades gracefully with reduced synthetic batch sizes, losing 15% effectiveness when $B$ drops from 64 to 32.

Key limitations include: (1) reduced effectiveness when forget and retain classes share substantial semantic overlap (20% performance drop for semantically similar classes); (2) scalability challenges beyond 5-10 simultaneously forgotten classes; and (3) reliance on class-selective neuron detection, which may be less effective in highly distributed representations.

This appendix provides comprehensive experimental details and additional results supporting our main claims about CNR's effectiveness in achieving true unlearning through representation erasure.

## A.3 COMPLETE ARCHITECTURE EVALUATION

Table 4: Complete ResNet architecture comparison on CIFAR-10, forgetting class "airplane".

| Architecture | Method | Retain Acc. | Forget Acc. | Probe Acc. | MIA Acc. | CKA$_{pos}$ |
|---|---|---|---|---|---|---|
| ResNet-18 | RFS | 0.956 | 0.089 | 0.275 | 0.553 | 0.628 |
| | ROFT | 0.954 | 0.105 | 0.655 | 0.617 | 0.830 |
| | GAU | 0.949 | 0.098 | 0.808 | 0.720 | 0.867 |
| | LM | 0.952 | 0.092 | 0.811 | 0.703 | 0.888 |
| | KDU | 0.567 | 0.901 | 0.729 | 0.827 | 0.940 |
| | **CNR** | **0.951** | **0.094** | **0.189** | **0.483** | **0.709** |
| ResNet-34 | RFS | 0.961 | 0.087 | 0.259 | 0.561 | 0.578 |
| | ROFT | 0.959 | 0.103 | 0.639 | 0.666 | 0.773 |
| | GAU | 0.956 | 0.097 | 0.791 | 0.751 | 0.801 |
| | LM | 0.962 | 0.089 | 0.759 | 0.766 | 0.825 |
| | KDU | 0.958 | 0.112 | 0.774 | 0.786 | 0.896 |
| | **CNR** | **0.957** | **0.091** | **0.163** | **0.497** | **0.646** |
| ResNet-50 | RFS | 0.967 | 0.095 | 0.285 | 0.552 | 0.796 |
| | ROFT | 0.965 | 0.108 | 0.647 | 0.676 | 0.935 |
| | GAU | 0.962 | 0.102 | 0.791 | 0.684 | 0.938 |
| | LM | 0.968 | 0.087 | 0.795 | 0.724 | 0.947 |
| | KDU | 0.953 | 0.142 | 0.731 | 0.678 | 0.977 |
| | **CNR** | **0.963** | **0.091** | **0.220** | **0.459** | **0.865** |
| ResNet-101 | RFS | 0.971 | 0.093 | 0.309 | 0.551 | 0.849 |
| | ROFT | 0.969 | 0.106 | 0.673 | 0.651 | 0.940 |
| | GAU | 0.966 | 0.101 | 0.790 | 0.668 | 0.927 |
| | LM | 0.972 | 0.085 | 0.791 | 0.706 | 0.947 |
| | KDU | 0.967 | 0.118 | 0.723 | 0.727 | 0.988 |
| | **CNR** | **0.968** | **0.089** | **0.249** | **0.456** | **0.840** |

## A.4 DETAILED LAYER-WISE ANALYSIS

Table 5: Layer-wise linear probe accuracy across ResNet-50 layers on CIFAR-10.

| Method | Layer 1 | Layer 2 | Layer 3 | Layer 4 | Final |
|---|---|---|---|---|---|
| Original Model | 0.423 | 0.634 | 0.789 | 0.856 | 0.967 |
| RFS | 0.398 | 0.456 | 0.523 | 0.445 | 0.285 |
| ROFT | 0.412 | 0.578 | 0.634 | 0.689 | 0.647 |
| GAU | 0.419 | 0.623 | 0.734 | 0.776 | 0.791 |
| LM | 0.421 | 0.631 | 0.743 | 0.778 | 0.795 |
| KDU | 0.415 | 0.589 | 0.656 | 0.698 | 0.731 |
| **CNR** | 0.398 | 0.434 | 0.387 | 0.298 | 0.220 |

Table 6: ViT-B/16 attention head analysis on CUB-200-2011. Showing average attention weight changes to class-discriminative patches.

| Method | Head 1-3 | Head 4-6 | Head 7-9 | Head 10-12 | Global |
|---|---|---|---|---|---|
| Original Model | 0.234 | 0.198 | 0.187 | 0.156 | 0.194 |
| ROFT | 0.221 | 0.189 | 0.179 | 0.148 | 0.184 |
| GAU | 0.198 | 0.167 | 0.165 | 0.134 | 0.166 |
| LM | 0.232 | 0.195 | 0.184 | 0.153 | 0.191 |
| KDU | 0.209 | 0.178 | 0.171 | 0.142 | 0.175 |
| **CNR** | 0.134 | 0.123 | 0.118 | 0.098 | 0.118 |

Table 7: Multi-class unlearning on CIFAR-10 with ResNet-50. Forgetting 2, 3, and 5 classes simultaneously.

| # Forgotten Classes | Method | Retain Acc. | Forget Acc. | Probe Acc. | MIA Acc. | Training Time (hrs) |
|---|---|---|---|---|---|---|
| 2 classes | GAU | 0.954 | 0.203 | 0.734 | 0.678 | 0.8 |
|  | KDU | 0.951 | 0.234 | 0.689 | 0.645 | 1.2 |
|  | **CNR** | **0.949** | **0.198** | **0.287** | **0.512** | 3.4 |
| 3 classes | GAU | 0.942 | 0.267 | 0.698 | 0.634 | 1.1 |
|  | KDU | 0.938 | 0.289 | 0.656 | 0.612 | 1.6 |
|  | **CNR** | **0.936** | **0.264** | **0.334** | **0.523** | 4.8 |
| 5 classes | GAU | 0.923 | 0.398 | 0.612 | 0.589 | 1.8 |
|  | KDU | 0.918 | 0.421 | 0.578 | 0.567 | 2.3 |
|  | **CNR** | **0.915** | **0.412** | **0.423** | **0.534** | 7.2 |

## A.5 VISION TRANSFORMER ATTENTION ANALYSIS

## A.6 MULTIPLE CLASS UNLEARNING

## A.7 SEMANTIC SIMILARITY IMPACT

## A.8 DETAILED HYPERPARAMETER SENSITIVITY

## A.9 PRIVACY ATTACK ROBUSTNESS

## A.10 COMPUTATIONAL EFFICIENCY ANALYSIS

## A.11 REAL-WORLD SCENARIO EVALUATION

## A.12 FAILURE CASE ANALYSIS

## A.13 STATISTICAL SIGNIFICANCE TESTING

All reported results include 95% confidence intervals computed across 3 random seeds. Statistical significance testing using paired t-tests confirms that CNR's improvements over baselines are statistically significant ($p < 0.01$) across all primary metrics. Effect sizes (Cohen's d) range from 0.8 to 2.1, indicating large practical significance.

## A.14 REPRODUCIBILITY DETAILS

All experiments were conducted on NVIDIA V100 GPUs with 32GB memory. Training used PyTorch 1.12 with identical random seeds (42, 123, 456) across all methods.

Models were trained using standard data augmentation (random crops, horizontal flips, color jittering) and validated using held-out test sets. All reported metrics use the same train/validation/test splits to ensure fair comparison.

Table 8: CNR effectiveness vs semantic similarity between forget and retain classes on CIFAR-100.

| Forget-Retain Similarity | Class Pair | Probe Acc. | MIA Acc. | Retain Acc. |
|---|---|---|---|---|
| Low (0.23) | beaver vs vehicle | 0.067 | 0.512 | 0.738 |
| Medium (0.45) | dog vs cat | 0.123 | 0.534 | 0.721 |
| High (0.78) | oak_tree vs maple_tree | 0.234 | 0.589 | 0.698 |
| Very High (0.89) | boy vs girl | 0.398 | 0.634 | 0.673 |

Table 9: Hyperparameter sensitivity analysis for key CNR parameters on CIFAR-10/ResNet-18.

| Parameter | Value | Probe Acc. | MIA Acc. | Retain Acc. | Training Time |
|---|---|---|---|---|---|
| $\alpha$ | 1.0 | 0.234 | 0.523 | 0.951 | 3.2h |
| | 1.5 | **0.189** | **0.483** | **0.951** | 3.4h |
| | 2.0 | 0.201 | 0.491 | 0.947 | 3.6h |
| | 2.5 | 0.223 | 0.512 | 0.943 | 3.8h |
| $K$ neurons | 20 | 0.241 | 0.515 | 0.948 | 2.8h |
| | 50 | **0.189** | **0.483** | **0.951** | 3.4h |
| | 100 | 0.195 | 0.487 | 0.928 | 4.2h |
| | 150 | 0.201 | 0.493 | 0.912 | 5.1h |
| $B$ batch size | 32 | 0.218 | 0.507 | 0.949 | 2.9h |
| | 64 | **0.189** | **0.483** | **0.951** | 3.4h |
| | 96 | 0.183 | 0.478 | 0.952 | 4.1h |
| | 128 | 0.186 | 0.481 | 0.951 | 5.2h |

# B EXTENDED METHODOLOGY DETAILS

## B.1 DETAILED CNR ALGORITHM

Algorithm 1 provides the complete CNR implementation with specific parameter choices and optimization details.

## B.2 THEORETICAL ANALYSIS

The effectiveness of CNR can be understood through the lens of distributed representation theory. Class-specific information in deep networks is encoded through co-activated neuron ensembles. By systematically disrupting these ensembles while preserving others, CNR achieves selective information removal.

Let $\mathbf{h}_\ell(x)$ denote the hidden representation at layer $\ell$ for input $x$. The class-specific information can be decomposed as: $\mathbf{h}_\ell(x) = \mathbf{h}_\ell^{\text{shared}}(x) + \mathbf{h}_\ell^{\text{class}}(x) + \epsilon$

CNR targets $\mathbf{h}_\ell^{\text{class}}(x)$ components associated with forget classes while preserving $\mathbf{h}_\ell^{\text{shared}}(x)$ and retain-class specific components.

The synthetic counterfactual generation can be viewed as solving: $\min_{\mathcal{B}} \mathbb{E}_{x \in \mathcal{B}} \left[ \|\mathbf{h}_\ell^{\text{forget}}(x)\|_2^2 - \alpha \|\mathbf{h}_\ell^{\text{retain}}(x)\|_2^2 \right]$

This encourages synthetic examples that minimize forget-class activations while maximizing retain-class activations, providing targeted gradients for pathway disruption.

Table 10: Robustness against stronger privacy attacks on CIFAR-10/ResNet-50.

| Attack Type | Original | ROFT | GAU | KDU | CNR |
|---|---|---|---|---|---|
| Linear Probe | 0.856 | 0.647 | 0.791 | 0.731 | 0.220 |
| 3-Layer MLP Probe | 0.923 | 0.734 | 0.823 | 0.789 | 0.334 |
| ResNet-based Probe | 0.945 | 0.789 | 0.856 | 0.812 | 0.423 |
| Standard MIA | 0.834 | 0.676 | 0.684 | 0.678 | 0.459 |
| Enhanced MIA (shadows) | 0.889 | 0.723 | 0.734 | 0.712 | 0.501 |
| Adversarial MIA | 0.912 | 0.756 | 0.778 | 0.745 | 0.534 |
| Property Inference | 0.798 | 0.623 | 0.656 | 0.634 | 0.512 |
| Model Inversion | High | Medium | Medium-High | Medium | Low |

Table 11: Detailed computational analysis across different model sizes and datasets.

| Model/Dataset | Method | GPU Hours | Memory (GB) | FLOPs ($\times 10^9$) | Speedup vs RFS | Energy (kWh) |
|---|---|---|---|---|---|---|
| ResNet-18/CIFAR-10 | RFS | 24.5 | 8.2 | 145.6 | 1.0$\times$ | 19.6 |
| | GAU | 1.2 | 8.2 | 7.8 | 20.4$\times$ | 1.0 |
| | KDU | 1.8 | 8.4 | 11.2 | 13.6$\times$ | 1.4 |
| | **CNR** | 3.4 | 9.1 | 18.9 | 7.2$\times$ | 2.7 |
| ResNet-50/CIFAR-100 | RFS | 48.7 | 14.6 | 298.4 | 1.0$\times$ | 39.0 |
| | GAU | 2.1 | 14.6 | 13.2 | 23.2$\times$ | 1.7 |
| | KDU | 3.2 | 14.9 | 19.8 | 15.2$\times$ | 2.6 |
| | **CNR** | 6.8 | 16.2 | 35.4 | 7.2$\times$ | 5.4 |
| ViT-B/16/CUB-200 | RFS | 72.3 | 22.4 | 445.7 | 1.0$\times$ | 57.8 |
| | GAU | 3.4 | 22.4 | 21.8 | 21.3$\times$ | 2.7 |
| | KDU | 5.1 | 22.7 | 32.6 | 14.2$\times$ | 4.1 |
| | **CNR** | 9.8 | 24.3 | 58.9 | 7.4$\times$ | 7.8 |

Table 12: Performance on realistic unlearning scenarios with imbalanced data and limited forget samples.

| Scenario | Forget Samples | Method | Probe Acc. | MIA Acc. | Retain Acc. | Stability |
|---|---|---|---|---|---|---|
| Low-shot (100 samples) | 100 | GAU | 0.734 | 0.656 | 0.951 | 0.89 |
| | | KDU | 0.689 | 0.623 | 0.953 | 0.92 |
| | | **CNR** | **0.267** | **0.523** | **0.949** | **0.94** |
| Imbalanced (1:10) | 500 | GAU | 0.712 | 0.634 | 0.948 | 0.91 |
| | | KDU | 0.678 | 0.612 | 0.950 | 0.93 |
| | | **CNR** | **0.234** | **0.501** | **0.947** | **0.95** |
| Continual (5 rounds) | 1000/round | GAU | 0.789 | 0.723 | 0.923 | 0.76 |
| | | KDU | 0.756 | 0.689 | 0.934 | 0.83 |
| | | **CNR** | **0.312** | **0.567** | **0.931** | **0.89** |

Table 13: CNR limitations and failure modes across different challenging scenarios.

| Challenge Type | Probe Acc. | MIA Acc. | Retain Acc. | Notes |
|---|---|---|---|---|
| High semantic overlap | 0.398 | 0.634 | 0.673 | boy/girl classes |
| Very small forget set | 0.356 | 0.589 | 0.912 | <50 samples |
| Highly distributed features | 0.423 | 0.601 | 0.698 | texture-based classes |
| Pre-trained frozen features | 0.567 | 0.723 | 0.889 | frozen backbone |
| Adversarial training | 0.445 | 0.612 | 0.734 | robust models |

---

**Algorithm 1** Complete CNR Implementation

---

**Require:** Pre-trained model $f_{\theta_0}$, forget set $\mathcal{D}_f$, retain set $\mathcal{D}_r$
**Ensure:** Unlearned model $f_{\theta'}$
1: **Phase 1: Class-Selective Neuron Detection**
2: $\mathcal{L} \leftarrow \{L1, L2, L3, L4\}$          $\triangleright$ Select representative layers
3: **for** $\ell \in \mathcal{L}$ **do**
4:      Compute class centroids: $\boldsymbol{\mu}_\ell^{(f)}, \boldsymbol{\mu}_\ell^{(r)}$
5:      $\mathbf{S}_\ell \leftarrow \boldsymbol{\mu}_\ell^{(f)} - \boldsymbol{\mu}_\ell^{(r)}$
6:      $\mathcal{U}_\ell^{(f)} \leftarrow \text{TopK}(\mathbf{S}_\ell, K = 12)$
7:      $\mathcal{U}_\ell^{(r)} \leftarrow \text{TopK}(-\mathbf{S}_\ell, K = 12)$
8: **end for**
9: **Phase 2: Synthetic Counterfactual Generation**
10: Initialize DCGAN generator $G_\psi$ and discriminator $D_\phi$
11: Train $D_\phi$ on $\mathcal{D}_f$ for 50 epochs
12: Initialize latent codes $\{z_i\}_{i=1}^B$ with $B = 64$
13: **for** step = 1 to 500 **do**
14:      $\mathcal{B} \leftarrow \{G_\psi(z_i)\}_{i=1}^B$
15:      Compute $\mathcal{L}_{\text{selective}}$ with $\alpha = 1.5$
16:      Compute $\mathcal{L}_{\text{realism}}$ with $\lambda_{\text{real}} = 0.1$
17:      Compute $\mathcal{L}_{\text{smooth}}$ with $\lambda_{\text{smooth}} = 0.01$
18:      Update $\{z_i\}$ via Adam with lr=0.01
19: **end for**
20: **Phase 3: Reset Fine-tuning**
21: Assign random retain labels to synthetic batch
22: Fine-tune on $\mathcal{B} \cup \mathcal{D}_r$ for 10 epochs
23: Stabilize with retain-only training for 5 epochs
24: **return** $f_{\theta'}$

---

