# OpenReview forum: "Beyond Masking and Avoidance: Toward True Unlearning"
_ICLR.cc/2026/Conference — ICLR 2026 Conference Desk Rejected Submission_

### Official Review · Reviewer_HMvD · 2025-10-22

**Soundness:** 2
**Presentation:** 1
**Contribution:** 2
**Rating:** 2
**Confidence:** 3

**Summary:**

The study introduces a three-stage unlearning pipeline, CNR, to improve the performance of current unlearning methods. Experiments on MNIST, CIFAR-10/100, LFW, and CUB-200-2011 with ResNet-18/34/50/101 and ViT-B/16 claim that CNR drives linear probes, k-NN, SVM, and membership-inference attacks to chance level while preserving > 95% retained accuracy.

**Strengths:**

1. The idea is straightforward and easy to understand.
2. Many layer-wise visualizations are provided. This can give some intuitive insight of the proposed method.

**Weaknesses:**

1. The paper writing is poor and hard to understand. For example, numerous undefined symbols (???) remain, the conclusion section is absent, and there is a missing proper reference to the baseline model. These may not be suitable for the formal submission standards.
2. CNR relies on layer-by-layer neuron selection and GAN training. The computational overhead grows linearly with the number of neurons and layers, especially on large models.
3. The effectiveness of CNR is only verified on the small-scale CNN and ViT. It is better to conduct the experiments on LLMs.
4. More ablation studies are needed to justify the selection of the hyperparameters, like K, $\alpha$, and $\lambda$.
5. The technical contributions are limited, since most of the three strategies are original from others, i.e., GAN for counterfactual generation and class-selective neurons.
6. The original code is not available to check.

**Questions:**

Pleace check the Weaknesses part.

---

### Official Review · Reviewer_Jw5n · 2025-10-26

**Soundness:** 2
**Presentation:** 1
**Contribution:** 3
**Rating:** 4
**Confidence:** 3

**Summary:**

This manuscript introduces an architecture agnostic method called Class-Specific Neuron Reset (CNR) for selectively unlearning class specific information. CNR achieves this via three stages:
- Identifying class-selective neurons via their mean activations.
- Resetting those neurons by fine-tuning on DCGAN-generated samples of the class to be unlearnt.
- Retaining overall performance via regularized supervised fine-tuning.
Authors have conducted various experiments and ablations on MNIST, CIFAR10, and CUB200 and their results suggest that CNR erases undesirable information while maintaining the performance.

**Strengths:**

Overall, I believe this paper is a gradient step in the right direction and while the idea is not novel per se, all the components together are novel enough and hopefully interesting for the ICLR community.
I particularly liked:
- comprehensive experiments and,
- the strong related work section

**Weaknesses:**

I think the manuscript lacks strong discussion. There are many cases that numerical results are reported in a table but there is no discussion e.g., why should as a reader I care or what is the conclusion from authors’ point of view? I strongly encourage authors to add this for all reported results.

The paper seems to be written in rush and the quality on multiple fronts should be improved significantly:
- In Figure 2 and Figure 3 labels, legends, and captions can be improved, the current size is too small

- Numerous instances of “?” across the manuscript which made understanding references and as a result the manuscript challenging: please revisit lines 215, 246, 258, 286, 307, 310, 314, 359, 365

- Equations are sloppy in terms of defining terms and sub/super scripts

- A5 to A12 sections are missing from the appendix.

Citations are missing for the selected baseline methods, in other words, authors nicely presented related work on this topic but I am not sure why none of these methods made it to the experiments for comparisons. Either citations are missing or the justifications for excluding them from comparisons.

While the manuscript has detailed report on the hyper-parameter values, the code seems to be missing. I urge authors to provide all the codes as well for a better reproducibility.

The driving motivation of this work is for foundation models but the experimental results tend to be on a smaller ViTs. It would make the message stronger if authors provide results on larger ViTs as well

I am happy to see that authors repeated each experiment 3 times with 3 different seeds and performed significant testing, assuming the reported numbers are average (it is missing in the manuscript), please provide standard error or confidence intervals as well (A13 section claims that all results are with confidence interval but I could not find any).
On this topic I have two questions:
- have you done p-value correction for multiple hypothesis testing? if so, what method was used?
- t-test requires a normality assumption, how did you ensure that this assumption is not violated?

**Questions:**

Question:

In real world scenarios, sometimes we only need to forget a subset of examples from a class as opposed to completely removing that class, how does CNR work for such cases? feel free to provide a discussion why it is not feasible or is possible to be clear providing empirical results is optional.

Score:

I would be happy to raise my score if authors address weaknesses: the presentation, missing discussions, and clarifying raised issues: reported numbers and significance testing, selected baselines, larger models experiment (least important one)

---

### Official Review · Reviewer_DWuX · 2025-10-29

**Soundness:** 3
**Presentation:** 1
**Contribution:** 3
**Rating:** 2
**Confidence:** 4

**Summary:**

In the current paper, the authors introduce Class-Specific Neuron Reset (CNR), a new approach for machine unlearning that aims to remove class-specific information from a model’s internal representations rather than merely suppressing its outputs. CNR operates in three stages: (i) identifying neurons highly selective to the forgotten class, (ii) generating synthetic counterfactual examples that deactivate these neurons while preserving retain-class features, and (iii) fine-tuning the model to reset targeted pathways without harming global performance. The method is evaluated on several datasets (MNIST, CIFAR-10/100, LFW, CUB-200) and architectures (ResNets, ViTs), showing that it achieves near-random decoding accuracy for forgotten classes while maintaining strong performance on retained ones.

**Strengths:**

These are the paper's strengths:
- It addresses an important and timely problem related to data privacy, model editing, and the right to be forgotten.
- It introduces a novel concept of “true unlearning”, shifting focus from output suppression to the actual removal of class-specific information from internal representations
- The proposed Class-Specific Neuron Reset (CNR) approach is architecture-agnostic, interpretable, and consists of three stages (detection, synthesis, reset)
- Provides comprehensive experimental validation across multiple datasets and architectures, with consistent improvements in representation-level erasure and minimal performance loss on retained classes.

**Weaknesses:**

These are the paper's weaknesses:
- The writing and presentation quality are poor: inconsistent notation, missing references (marked “???”), and lack of a proper conclusion section reduce clarity and professionalism.
-  The theoretical foundations of representation erasure are underdeveloped. Claims are mainly supported by empirical evidence without formal guarantees or analysis of generalization.
- The Related Work section is fragmented and uneven, containing outdated citations and missing recent or relevant ones.
- Effectiveness of the proposed approach degrades when forget and retain classes are semantically similar, indicating limitations in disentangling overlapping representations.

**Questions:**

Besides the weaknesses mentioned above, here is the rest of my concerns:
- The paper defines “true unlearning” as the state in which no simple or robust decoder can recover forgotten information from the hidden representations. While this idea is intuitively clear, can you present it in a more formal manner, e.g. formulating it as an optimization objective?
- The neuron selectivity metric is based on mean activation differences. While this mean-difference metric is simple and intuitive, it is also very limited because it only captures average activation differences and ignores other important aspects. Could you provide a justification why did you choose this one, instead of other more statistically relevant metrics such as: Fisher Discriminant score or Mutual Informaiton?
-  In the synthetic counterfactual generation, how can you ensure that minimizing the $L_{selective}$ loss truly removes forget-class information rather than just redistributing it?
- How are the hyperparameters $\lambda_{real}$, $\lambda_{smooth}$ chosen? Can you present a parameter sensitivity analysis?
- In multi-class forgetting, how does the algorithm handle overlapping neuron selectivity among different forget classes?
- What if instead of forgetting a whole class, I would like to forget specific instances of a class. How will your method should be changed in this case?

---

### Official Review · Reviewer_sQEr · 2025-11-01

**Soundness:** 2
**Presentation:** 1
**Contribution:** 2
**Rating:** 2
**Confidence:** 3

**Summary:**

This work is motivated by the claim that current unlearning methods do not eliminate class-specific information from internal representations and merely 'avoid' predicting a particular class. To tackle this challenge, the paper presents a class unlearning algorithm, **CNR**, which finetunes the network using synthetic samples generated by a GAN. It also presents a set of metrics to demonstrate superior performance compared to baselines.

**Strengths:**

**Sufficient Experimental Slate**: The paper measures the success of the proposed approach with several relevant metrics

**Weaknesses:**

**Technical content unclear**:
The paper proposes a new loss function to perform unlearning, but the theoretical motivation (or even intuition) for the proposed losses is not clear. Appendix B.2 claims to provide a theoretical analysis but makes no claims in its content.

**Poor Motivation**:
Several papers cited in this work to motivate the need for representational erasure are based on results for LLMs (See Shi et al, Zhang 2024a, Wang 2024a, Lynch 2023). It is unclear why representation erasure is key to unlearning in the vision classification setting addressed in this work.

**Poor Experimental Design**:
Experiments seem to be conducted only only forgetting a single arbitrary class. This is not in line with the premise of the work, which places a high importance on preserving retained class representations, since there is no experiment to indicate the outcome of the proposed algorithm when representations of classes are similar.
For example, unlearning the class ''cars'' in a classification model trained on CIFAR10 may have a disproportionate impact on ''trucks'' since the representations of these classes may be similar, especially in early layers.

**Poor Scholarship and Bad Typsetting**:
There are several examples of poor scholarship throughout the paper.
- There are several missing citations
- The conclusions section of the paper is provided in Appendix A without context.
- Several sections in the appendices are empty (A.5 to A.12)
- Appendix A.13 indicates that the paper reports 95% confidence intervals, but does not seem to provide them in any of the tables or figures in the main body or in the appendix.

There are several instances of bad typesetting, mostly involving incorrect use of subscripts and superscripts. Please use something like \(a_{b_{c}}^{d_{e}}\) to correctly render nested subscripts and superscripts.

**Missing key references/Unclear references**:
Some key references in the Bibliography do not seem to exist. For example, Jang et al’s ''Unlearning in deep neural networks'' does not seem to appear in the proceedings of NeurIPS 2022 or 2023. Similarly, Patil et al.’s ''Attacking safety alignment and unlearning in open-source llms'' points to an invalid arxiv link, and I was unable to find a paper by this author.

For these reasons, the recommendation for the paper is not higher and thus must recommend a reject.

**Questions:**

1. The authors state the term ''true unlearning'' differently on separate occasions: In Lines 013-014, they state ''define true unlearning as the elimination of class-specific information from hidden states such that no simple or robust decoder can recover it'' and in Lines 042-044, they state ''By true unlearning we mean that the post-unlearning model behaves like a model retrained from scratch on data that exclude the forget subset''. Neither statement is necessarily implied by the other. Can the authors clarify?
2. The authors benchmark against ''chance level performance'' when unlearning. However, Several works, Jia et al., Murti et al., compare against models trained on only the retain set, which obtain 0% accuracy on the forget set. Can the authors clarify what the benchmark for performance is?
3. The paper refers to Boyd and Vandenberghe 2004 and Rockafellar 1997 for unclear reasons. Can the authors provide justifications for this citation in their related work?
4. This paper addresses only class unlearning. How would the proposed method provide insights into random forgetting, where the representations of the forget samples need not be distinct from the remain samples?
5. The term Centered Kernel Alignment (CKA) is referenced throughout the text, but is neither defined nor is a relevant citation provided. Could the authors clearly define or provide a relevant citation?
6. The code used in this work has not been made public, raising concerns regarding reproducibility. Do the authors intend to release the code?

Looking forward to open discussions with the authors to help improve the work.

### References
---
J. Jia, J. Liu, P. Ram, Y. Yao, G. Liu, Y. Liu, P. Sharma, and S. Liu. Model sparsity can simplify
machine unlearning. In Thirty-seventh Conference on Neural Information Processing Systems,
2023.

C. Murti and C. Bhattacharyya. DISCEDIT: Model Editing by Identifying Discriminative Components. In Thirty-eighth Conference on Neural Information Processing Systems,
2024.

**Details Of Ethics Concerns:**

- This work cites a few papers as key references that do not seem to exist. For example, Jang et al. 2022 (cited on lines 81, 87, 125, 146) is claimed to be published in NeurIPS 2022, but no such paper can be found in the proceedings. Similarly, the citation for Patel et al. 2024 (cited in line 91) points to an invalid arxiv link, and I am unable to find such a paper.

- Appendix A.13 contains claims that all results include confidence intervals and mentions statistical tests that do not exist in either the main body or the supplementary.

---

### Official Review · Reviewer_SxA3 · 2025-11-01

**Soundness:** 2
**Presentation:** 1
**Contribution:** 1
**Rating:** 0
**Confidence:** 4

**Summary:**

This paper proposes a technique for unlearning, aiming to remove class-specific information from the neurons. Class-specific neurons are identified by measuring the differences between the mean features generated by the forget set and those generated by the remain set.  Synthetic data is then generated to 'down-activate' the neurons, which is then used to fine-tune the model along with the remain-set data.  Experiments are conducted on a variety of image classification tasks.

**Strengths:**

S1. The problem addressed in this paper is interesting and relevant in modern machine learning.

**Weaknesses:**

W1. The writing of the paper is extremely poor. In particular:
*  Several citations are for papers that do not exist (Jang et al, Li et al, Patil et al)
* There are numerous incorrect tags leading to (?), (??), and (???).
* Subscripts have not been used properly in LaTeX, rendering section 3 almost unreadable, and very difficult to parse. See, for instance, Equations 2, 4, 5, 6, and 7, as well as inline equations throughout section 3.

W2. The related works section also has potentially incorrect attributions. For instance, the line "Overlap between forget and retain semantics induces a tug-of-war: underforgetting
when the retain budget is tight, or collateral damage when it is loose (Ji et al., 2024; Zhang et al.,
2024d; Boyd and Vandenberghe, 2004; Rockafellar, 1997; Ehrgott, 2005; Fliege and Svaiter, 2000;
Lin et al., 2024; Pan et al., 2025; Patil et al., 2023; Dong et al., 2024)" miscites the books by Boyd and Vandenberghe as well as that of Rockafeller: these books do not address optimization-based unlearning *at all*, and it is incorrect to cite them here. Similarly, the paper by Cha et al, 2025, feels incorrectly attributed as well - the key conceit of that work is to use specially initialized LoRA adapters to effect unlearning in LLMs.

W3. While this work has cited a variety of LLM based unlearning papers, particularly as motivation, not a single LLM unlearning experiment has been cited. There are plenty of papers motivating unlearning (specifically, classwise unlearning as is the case in this work) that would have been more appropriate.

W4. In the Appendix, there are several empty subsection (Section A5-A12)

W5. Similarly, in the appendix, confidence intervals for experiments are promised, but are not stated in any of the subsequent tables.

W6. Moreover, several recent works (see, for instance, Kodge et al, 2024) that achieve perfect unlearning (forget set accuracy is 0), using representation based methods. This, and similar works, should be cited properly.

**Questions:**

Q1. To identify class-specific neurons, you measure the mean separation between the forget class features and the remain class features. Can the authors comment on why they chose this heuristic, which ignores the fact that some features may be close in the mean, but have significantly different variances (or higher order moments)?

**Details Of Ethics Concerns:**

It is clear that LLMs have been used *significantly* in the production of this work. There are several incorrect citations:

1. Xinyu Li et al. Representation-based unlearning in large language models. arXiv preprint
arXiv:2401.00000, 2024.

2. Anshuman Patil et al. Attacking safety alignment and unlearning in open-source llms. arXiv preprint
arXiv:2403.00000, 2024.

3. Hyeonwoo Jang et al. Unlearning in deep neural networks. In Proceedings of the 36th Conference
on Neural Information Processing Systems (NeurIPS), 202

As well as citations that are misspecified, i.e.
Dawen Zhang, Pamela Finckenberg-Broman, Thong Hoang, Shidong Pan, Zhenchang Xing, Mark
Staples, and Xiwei Xu. Right to be forgotten in the era of large language models: Implications,
challenges, and solutions, 2024b

Which should be cited as either a 2023 paper (arxiv version) or a 2025 paper (AI and Ethics springer monograph)

There are also a large number of broken citations/tags, bad LaTeX (subscripts not being used correctly), and mis-cited papers (see the way the books by (a) Boyd and Vandenberghe, and (b) Rockafeller have been cited).

---

### Note · Program_Chairs · 2026-01-17
**Submission Desk Rejected by Program Chairs**

The following references in this submission do not refer to real documents and/or have major errors in bibliographic information:

 Anshuman Patil et al. Attacking safety alignment and unlearning in open-source llms. arXiv preprint arXiv:2403.00000, 2024.